# Magnitude of consistent condom use and associated factors among people living with HIV/AIDS in Ethiopia: Implication for reducing infections and re-infection. A systematic review and meta-analysis

**Firomsa Bekele**[1]*, **Lalise Tafese**[2], **Teshome Debushe**[3]

1 School of Pharmacy, Institute of Health Science, Wallaga University, Nekemte, Ethiopia, 2 Department of Health Informatics, College of Health Science, Mattu University, Mattu, Ethiopia, 3 Department of Information Technology, College of Engineering and Technology, Mattu University, Mattu, Ethiopia

* firomsabekele21@gmail.com

**Data Availability Statement:** The data only available upon request. The data would be guarded carefully by our third party data for the only

## Abstract

### Background

The human immune virus or acquired immune deficiency syndrome, is a major threat to the health of millions of people worldwide. In Ethiopia, there were more than a million people living with HIV/AIDS. The continuous and appropriate use of condoms, particularly among those who have HIV-positive clients, is essential to a comprehensive and long-term approach to avoiding HIV and other STIs as well as unintended pregnancy. But irregular condom use is regularly reported. There is inconsistent reports of the prevalence and correlates of frequent condom use in Ethiopia. This study's goal is to provide an overview of the most recent research on magnitude condom use among people living with HIV in Ethiopia.

### Methods

Four databases of PubMed, Science Direct, Scopus, and Google Scholar were used. Finally, 10 studies that satisfied the eligibility criteria were included in the systematic review and meta-analysis. The data were collected using a methodical checklist for data extraction, and STATA 14 was utilized for the analysis. The consistent condom use was reported as use of condom in every sexual encounter preceding the study. The prevalence of consistent condom usage among HIV/AIDS patients was calculated by dividing the total number of patients who regularly used condoms by the total number of HIV/AIDS patients and multiplying that result by 100. The factors associated with a consistent use of condom were described using the pooled odds ratio (OR) and calculated based on binary outcomes from the included primary studies. The statistical significance was determined based on the correlation factor as their confidence level should not include 1. Subgroup analyses by region and publication years were carried out by using a random-effects model. The STATA commands of metan magnitude semagnitude, random xlab(.1,5,10) lcols (authors) by

purpose of this scientific study. Participants were not signed consent for data publicly. For all these reasons and following the indications of the research review committee of Institute of Health Sciences, Wallaga University, the authors must not upload the dataset to a stable, public repository. Interested, qualified researchers can access the data by requesting our third party, Wallaga university (Wu@ethionet.et)

**Funding:** The author(s) received no specific funding for this work.

**Competing interests:** The authors have declared that no competing interests exist.

**Abbreviations:** AIDS, Acquired Immune Deficiency Disease Syndrome; AOR, : Adjusted Odds Ratio; ART, : Anti-Retroviral Therapy; CI, Confidence Interval; HIV, : Human Immune Deficiency Virus; PLHIV, : Peoples Living with Human Immune Deficiency Virus; PMTCT, Prevention of Mother-To-Child HIV Transmission; PRISMA, : Preferred Reporting Items for Systematic Reviews and Meta-analyses; STI, Sexually Transmitting Infections.

(variables)texts(120) xsize(18) ysize (14) were used to carried out the subgroup analysis. To assess the presence of publication bias, funnel plot, Egger test and Begg's test at 5% significant level were computed. The asymmetry of funnel plot and the Egger test and Begg's test P value of 0 >0.5 showed the absence of publication bias. The Cochrane Q test statistic and $I^2$ tests were used to assess heterogeneity.

## Result

The pooled magnitude of consistent condom use was 50.56% (95%CI: 38.09–63.02). The predictors of consistent condom use includes urban residence (AOR = 3.46; 95% CI: 2.24–5.35), marital status (AOR = 0.33; 95% CI: 0.18–0.61), and HIV disclosure status (AOR = 5.61;95%CI: 2.29–13.73).

## Conclusion

Half of the HIV/AIDS patients in our study regularly used condoms. According to this study, urban residency, disclosure status, and marital status were all associated with consistent condom use among HIV/AIDS patients. Therefore, health education about condom use should be provided to married couples and people living in rural regions. In addition, disclosing HIV status and the necessity of constant condom usage would be crucial for consistent condom use.

## Background

Human immune virus and acquired immune deficiency syndrome (HIV/AIDS) affect millions of individuals worldwide [1, 2]. Worldwide, around 37.6 million people have HIV/AIDS in 2022 [1].

Sub-Saharan Africa is the area with the greatest impact, accounting for over two thirds of all new HIV infections worldwide in 2021. Most HIV-positive people live in low- and middle-income countries, including Ethiopia [2]. Despite a significant expansion of comprehensive HIV/AIDS interventions during the MDG (Millennium Development Goals) era, the HIV/AIDS burden in Ethiopia is still high [3]. Ethiopia saw a greater rate of new HIV infections in 2016 than it did in 2010 according to the 2017 version of the United Nations Programme on HIV/AIDS (UNAIDS) global AIDS report [4]. Data from the Ethiopia Demographic Health Survey (EDHS) show that more than a million Ethiopians have HIV/AIDS [5].

One of Ethiopia's Federal Ministry of Health's top health priorities is preventing new HIV infections. To do this, the Ethiopian government developed and put into practice a number of HIV prevention strategies, including expanding access to antiretroviral medication (ART) and a national condom distribution program [1].

The regular and appropriate use of condoms is essential to a comprehensive and long-term strategy for the prevention of HIV and other STIs, as well as for the prevention of unintended pregnancy [2, 6–8]. It is possible to prevent mother-to-child HIV transmission (PMTCT) by using condoms consistently during pregnancy. The use of condoms has been proven to be successful, with transmission rates being reduced by 80–95% [1, 9].

Regular condom usage and antiretroviral medication (ART) have decreased HIV disease mortality and morbidity, improving the health of people with HIV and allowing many of them to live longer and healthier lives [10]. Despite these efforts, the problem still affects the public

health, and the nature of the disease makes it difficult to reach the nation's goal for HIV/AIDS prevention and control [1].

Despite how important it is, regular condom use is underutilized. The problem primarily affects sero-discordant couples. The prevalence of inconsistent condom use is relatively high in Sub-Saharan Africa, particularly in Ethiopia [2]. The chance of the virus spreading and the generation of novel viral strains with treatment resistance has increased among HIV/AIDS patients on ART due to inconsistent condom use [10].

The prevalence of condom use varies among nations [1]. The reports of a seven-country community-based participatory research in the Asia-Pacific region revealed that a total of 57% of peoples living with HIV/AIDS practiced consistent condom use at sexual intercourse with their regular partner [11]. The study of South Africa found that 77% of the patents were consistently used condom [12].

The variable magnitudes were reported in SSA. Accordingly, in Nigeria(70.6%) [13], Egypt 45% [14], Kenya 65% [9], the prevalence of risky sexual behavior in Zambia was 71.1% [15], Mozambique(17%) [16]. The prevalence of Consistent condom use among sexually active HIV-positive women of Amhara region of Ethiopia was found to be 56.7% [17].

Numerous studies on the prevalence of condom use in Ethiopia and its causes have been published [1, 2, 6, 7, 10, 18–21]. Some of the things that are thought to be obstacles to the consistent use of condoms with HIV are the belief that condoms are not important in an HIV positive sero-concordant relationship, poor sexual satisfaction with condoms, the need to raise children, the husband's alcohol use, anxiety, depression, inadequate counseling by healthcare providers, low education, lower CD4 count, cost of condom, religious ideology, non-disclosure of HIV status, and shorter time on ART.

There hasn't been a systematic review or meta-analysis of the prevalence and predictors of consistent condom use in Ethiopia, and the previous comparable reviews and analyses have focused on risky sexual behavior, which is distinct from consistent condom use and includes having multiple sexual partners, initiating sex early, engaging in sex with commercial sex workers, and engaging in unprotected sex with a same-sex partner. Only the patterns of condom use are reported in our study. In order to offer a suitable intervention, the goal of this study is to synthesize the most recent research on the scope and contributing variables of regular condom use.

## Methods

### Searching strategy

The objective of the review was to determine the magnitude of consistent condom use and factors associated with consistent condom use among peoples living with HIV/AIDS in Ethiopia. The protocol of PRISMA 2020 was used to undertake this systematic review and meta-analysis [22]. Four data bases of PubMed, Science direct, Scopus and Google scholar were used. The time period used to conduct this review was from the May 2 to June 1, 2023. The last date to search was May 26, 2023. The MESH term for the database were ((Magnitude) OR (condoms) AND (Associated factors) AND (HIV/AIDS) AND (Ethiopia)). The Review protocol was registered on PROSPERO CRD42023430396

### Data collection process, items and extraction

Two authors, FB and LT, were responsible for gathering various works of literature. Utilizing endnote version X7.2, reference management software combines database search results and eliminates duplicate content. Two data extractors (FB and LT) used a standardized data extraction checklist on Microsoft Excel to extract the data. The data extraction checklist for the first

outcome (magnitude) comprised the author's name, the publication year, the area, the study design, the sample size, and the number of individuals who had the outcome. In order to determine the log OR for each component based on the results of the original research, data for the second outcome (related factors) were retrieved in the form of two by two tables. After discussion, disagreements between two independent reviewers were settled by bringing in a third reviewer (TD).

## Eligibility criteria

The findings published related to magnitude and predictors of consistent condom use among people living with HIV/AIDS in Ethiopia having all primary outcome and full texts available were included. The articles with unknown primary outcomes, systematic review and meta-analysis studies, preprint, short communications, and letter to the editors were excluded. The cross-sectional studies were included whereas the qualitative and non-observational studies were excluded. The review used the CoCoPop (condition, Context, and Population) framework to assess the eligibility of the studies. The study Population (POP) was HIV/AIDS patients, the Condition (CO) was consistent use of condom, and the context (CO) studies conducted in Ethiopia.

## Outcome measurement

There were basically two outcomes. The main outcome of interest was the prevalence of regular condom usage among HIV/AIDS patients, which was calculated by dividing the total number of patients who regularly used condoms by the total number of HIV/AIDS patients and multiplying that result by 100. The second outcome was identifying factors associated with a consistent use of condom, which were determined using the odds ratio (OR) and calculated based on binary outcomes from the included primary studies. The consistent condom use is operationalized as use of condom in every sexual encounter preceding the study [6, 20, 21].

## Quality assessment

The Joanna Briggs institute meta-analysis of statistics assessment and review instrument (JBI-MAStARI) was used for quality assessment [23]. The criteria used for assessing the quality of included studies were as follows; were the criteria for inclusion in the sample clearly defined?, were the study subjects and the setting described in detail?, was the exposure measured in a valid and reliable way?, were objective, standard criteria used for measurement of the condition?, were confounding factors identified?, were strategies to deal with confounding factors stated?, were the outcomes measured in a valid and reliable way?, and was appropriate statistical analysis used?. Using a quality assessment checklist, two authors (FB and LT) evaluated the article's quality. Third reviewers (TD) were brought in to settle disagreements between two independent reviewers after discussions for a potential consensus. Accordingly, the articles were classified as high quality if the score is >80%, moderate if (65–80%), and low if <65%.

## Data analysis and synthesis

In order to determine the pooled effect size with 95% CIs, data were exported to STATA V. 14. The Cochran Q test (chi-squared statistic) given as the p-value and $I^2$ statistic on forest plots were computed to examine heterogeneity among the included studies. At P 0.05, the Cochran's Q statistical heterogeneity test is deemed statistically significant. $I^2$ statistics range from 0% to 100%, and values of 0, 25, 50, and 75% were regarded as indicating no, minimal, low, and significant levels of heterogeneity, respectively. A random-effects model was used to estimate the

pooled magnitude because the first outcome showed signs of moderate heterogeneity. Two covariates (HIV disclosure status and marital status) were not found to be heterogeneous, hence a fixed-effects was employed. The high degree of heterogeneity was observed for three factors (level of education, ART duration, and counseling on condom use) hence, the random-effects model was used to estimate the Der Simonian and Laird's pooled effect. For the remaining one factor (urban residence) with low degree of heterogeneity, the random-effects model was employed to estimate the Der Simonian and Laird's pooled effect.

Subgroup analyses by region and publication years were carried out and sensitivity analysis was conducted by using a random-effects model. A funnel plot was used to assess publication bias. Asymmetry of the funnel plot is an indicator of publication bias. Besides, Egger's weighted regression and Begg's test were used to check publication bias. Statistical significance of publication bias was declared at a P-value of less than 0.05.

## Results

### Search results

A total of 5,435 articles were obtained up on initial searching from PubMed, Science direct, Scopus and Google scholar. A total of 4947 articles were removed due to duplications. Finally a total of 459 articles were excluded by observing their title and abstracts. Consequently, only 29 articles were subject to a full-text review. Finally, 10 articles were selected to be included in our review (Fig 1).

### Characteristics of included studies

In our systematic review and meta-analysis the filtered articles were cross-sectional studies. The majority of the participants were female in all articles [1, 2, 6, 7, 10, 18–21, 24]. The total sample was 4,219 HIV patients, ranging from 317 to 677. Regarding to the study settings, six articles were from Amhara [1, 2, 6, 7, 18, 20] and one each from Sidama [19], Tigray [21], Addis Ababa [24] and Benishan gul Gumuz [10] (Table 1).

### Quality assessment of included studies

In our systematic review and meta-analysis 7 articles [1, 2, 7, 10, 20, 21, 24] had a high quality and 3 articles [6, 18, 19] had moderate quality according to The Joanna Briggs institute meta-analysis of statistics assessment and review instrument (JBI- MAStARI) (Table 2).

### Magnitude of consistent condom use

The pooled magnitude of consistent condom use was 50.56% (95%CI: 38.09–63.02). Heterogeneity was observed across the included studies ($I^2$ = 54.9%, P = 0.018). Both the highest (78.9%) [6], and lowest (14.2%) [1] consistent use of condom was reported in Amhara regions (Fig 2). The moderate heterogeneity showed the presence of moderate variations across studies rather than by chance. This heterogeneity can alter the pooled magnitude estimate of consistent condom use. Therefore, subgroup analysis and sensitivity analysis was carried out. The region was the possible causes of this moderate heterogeneity. The possible reasons for heterogeneity, other than clinical differences, might include methodological issues such as problems with sampling technique.

### Subgroup analysis

The magnitude of consistent condom use of HIV/AIDS patients was computed based on the Publication years (Fig 3) and regions where studies were conducted (Fig 4). According to the

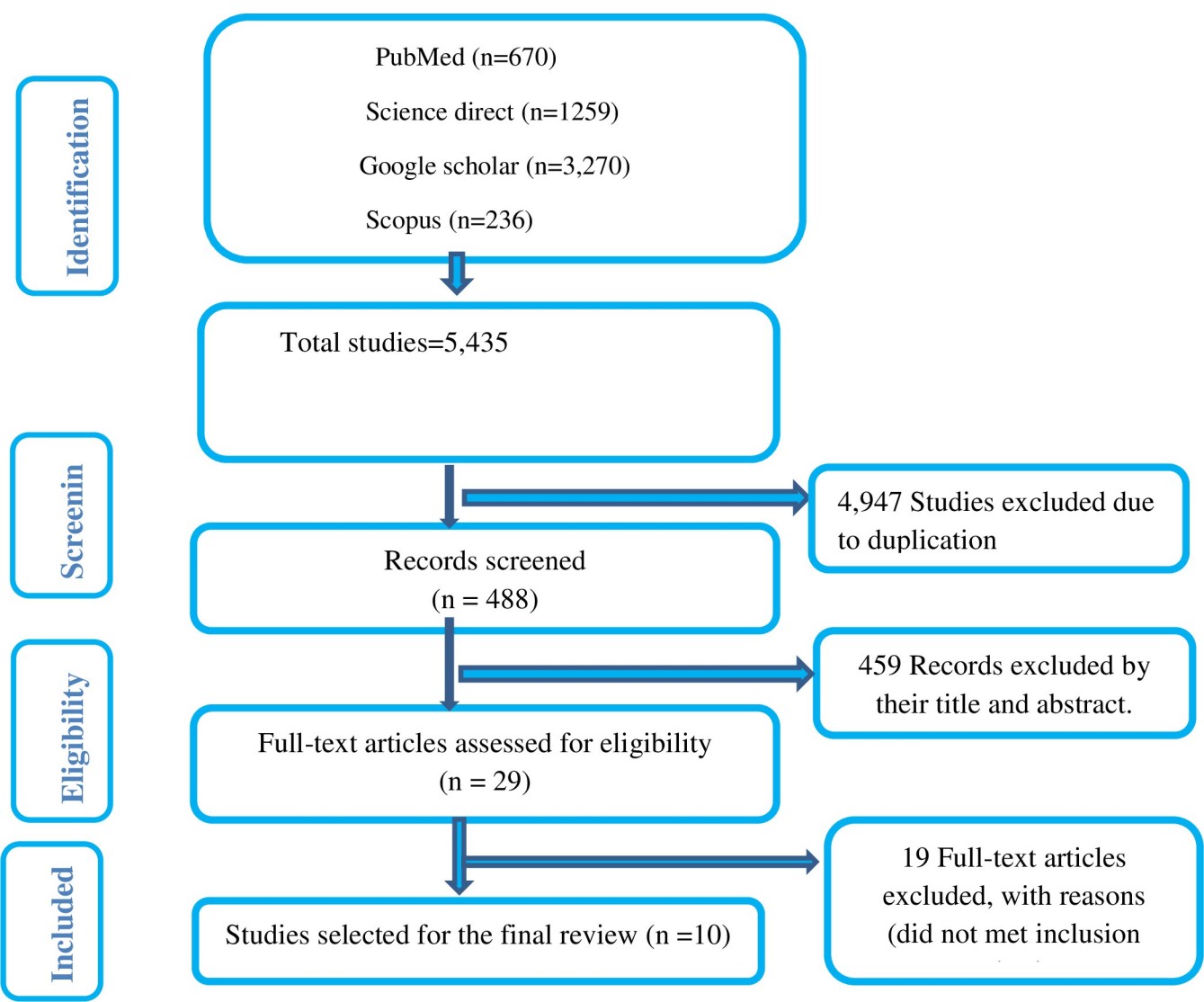

**Fig 1. Flow chart of the systematic research and study selection process.**

**Table 1. Summary of included studies on consistent condom use among HIV/AIDS patients in Ethiopia, 2023.**

| Authors | Years of publication | Region | Study design | Sample size | Gender(Female) | Consistent use (95%CI) |
|---|---|---|---|---|---|---|
| Wolde et al [2] | 2021 | Amhara | Cross-sectional | 401 | 54.6% | 58.4(53.1–63.1) |
| Nebiyu et al [1] | 2022 | Amhara | Cross-sectional | 423 | 100% | 14.2(10.9,17.5) |
| Estifanos et al [11] | 2012 | Amhara | Cross-sectional | 454 | 62.6% | 44.0(39.43,48.76) |
| Alex et al [10] | 2021 | Benishan gul Gumuz | Cross-sectional | 419 | 62.8% | 49.2%(42.2, 56.5) |
| Zewdneh et al [6] | 2015 | Amhara | Cross-sectional | 317 | 50.2% | 78.9%(73.95,83.23) |
| Mohammed et al [7] | 2019 | Amhara | Cross-sectional | 358 | 60.3% | 55.8%(50.55,61.08) |
| Biruk et al [19] | 2020 | Sidama | Cross-sectional | 358 | 58.1% | 51.4%(46.09,56.68) |
| Getie et al [20] | 2022 | Amhara | Cross-sectional | 400 | 100% | 59.1%(54.0,63.86) |
| Yemane et al [21] | 2015 | Tigray | Cross-sectional | 412 | 74.5% | 55.7%(50.64,60.45) |
| Rahel et al [24] | 2020 | Addis Ababa | Cross-sectional | 677 | 61.3% | 45.2%(41.40,49.04) |

**Table 2. Summary of quality assessment of included studies on consistent condom use among HIV/AIDS patients in Ethiopia, 2023.**

| Authors | Quality assessment |
| --- | --- |
| Wolde et al [2] | High |
| Nebiyu et al [1] | High |
| Estifanos et al [7] | Moderate |
| Alex et al [3] | High |
| Zewdneh et al [4] | Moderate |
| Mohammed et al [5] | High |
| Biruk et al [9] | Moderate |
| Getie et al [10] | High |
| Yemane et al [18] | High |
| Rahel et al [21] | High |

publication year, the magnitude of consistent condom use ranged from 14.2% (95%CI: -5.55–33.95) in 2022 [1] to 68.74% (95%CI: 46.18–91.30) in 2015 [6, 21]. However the heterogeneity was not found in terms of publication years. In relation to their geographical region the magnitude of consistent condom use ranged from 45.2% (95%CI: 17.04–73.36) in Addis Ababa [24] to 55.70(95%CI:27.59–83.81)in Tigray [21]. The heterogeneity was detected in Amhara region

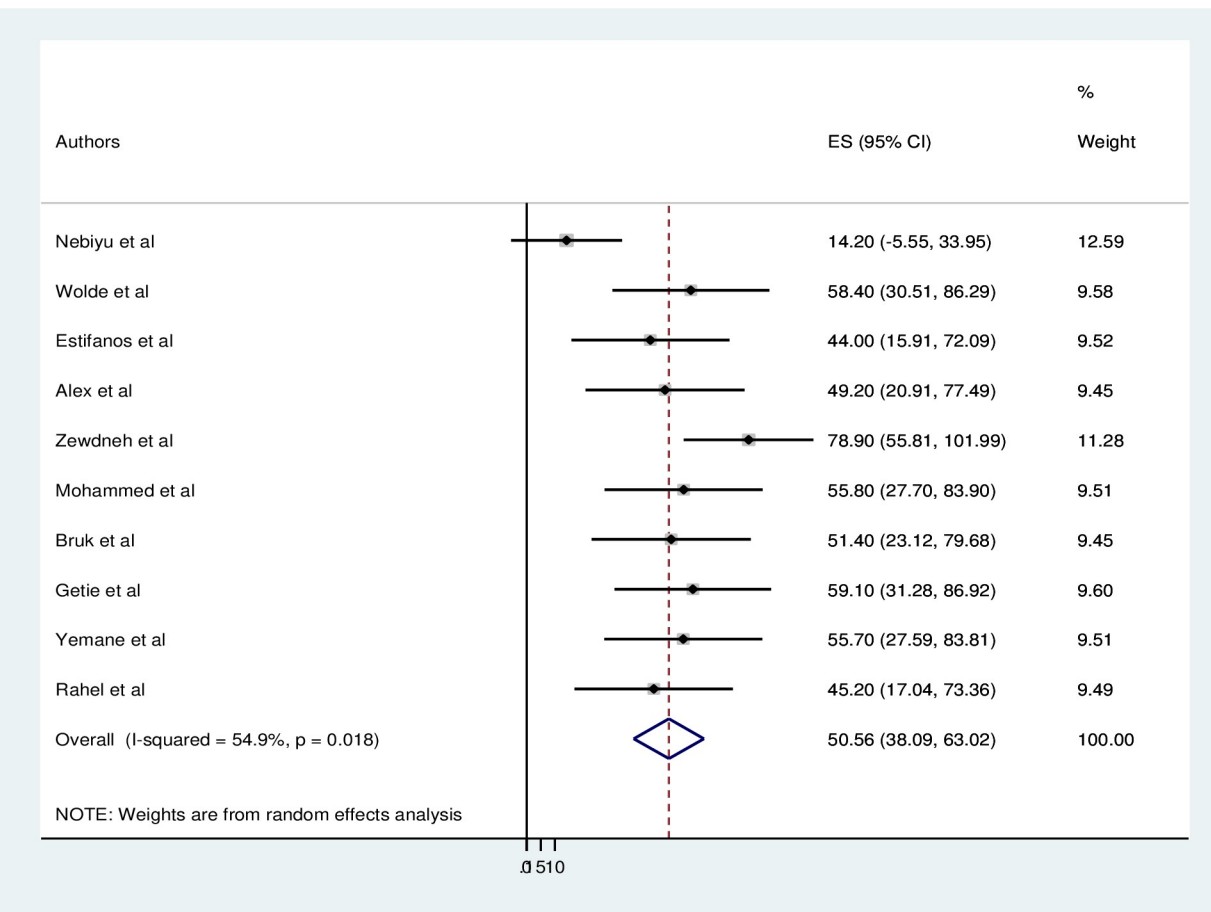

**Fig 2. Forest plot of the pooled magnitude of consistent use of condom among HIV/AIDS patients in Ethiopia, 2023.**

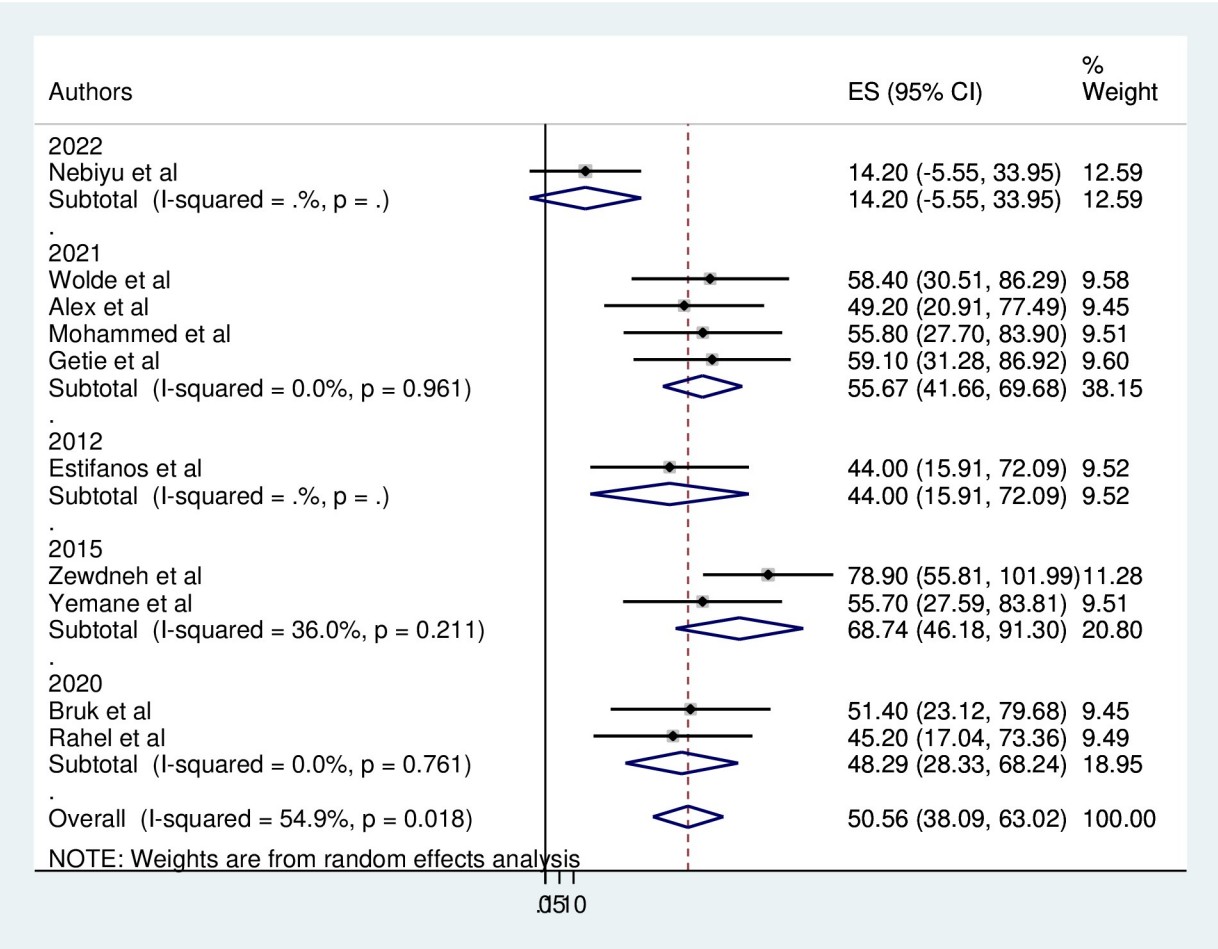

**Fig 3. Subgroup analysis on magnitude of consistent condom use among HIV/AIDS patients by their publication year in Ethiopia, 2023.**

($I^2$ = 74.6%, P = 0.001). The lowest and highest magnitude was reported 14.2% [1] and 78.90% [6], respectively in Amhara region.

## Sensitivity analysis

A random-effects model was used to conduct sensitivity analysis in order to determine whether a single research had an impact on the results of the overall meta-analysis. The results revealed that there was no strong support for this claim. The table shown that estimates from individual studies are more similar to combined estimates, suggesting that there is no single study effect on an entire study (Fig 5).

## Publication bias

To assess the presence of publication bias, funnel plot, Egger test and Begg's test at 5% significant level were computed. The funnel plot looks asymmetry, but the Egger test and Begg's test showed there is no statistically significant for the presence of publication bias with P-value = 0.275 and 0.405, respectively (Fig 6).

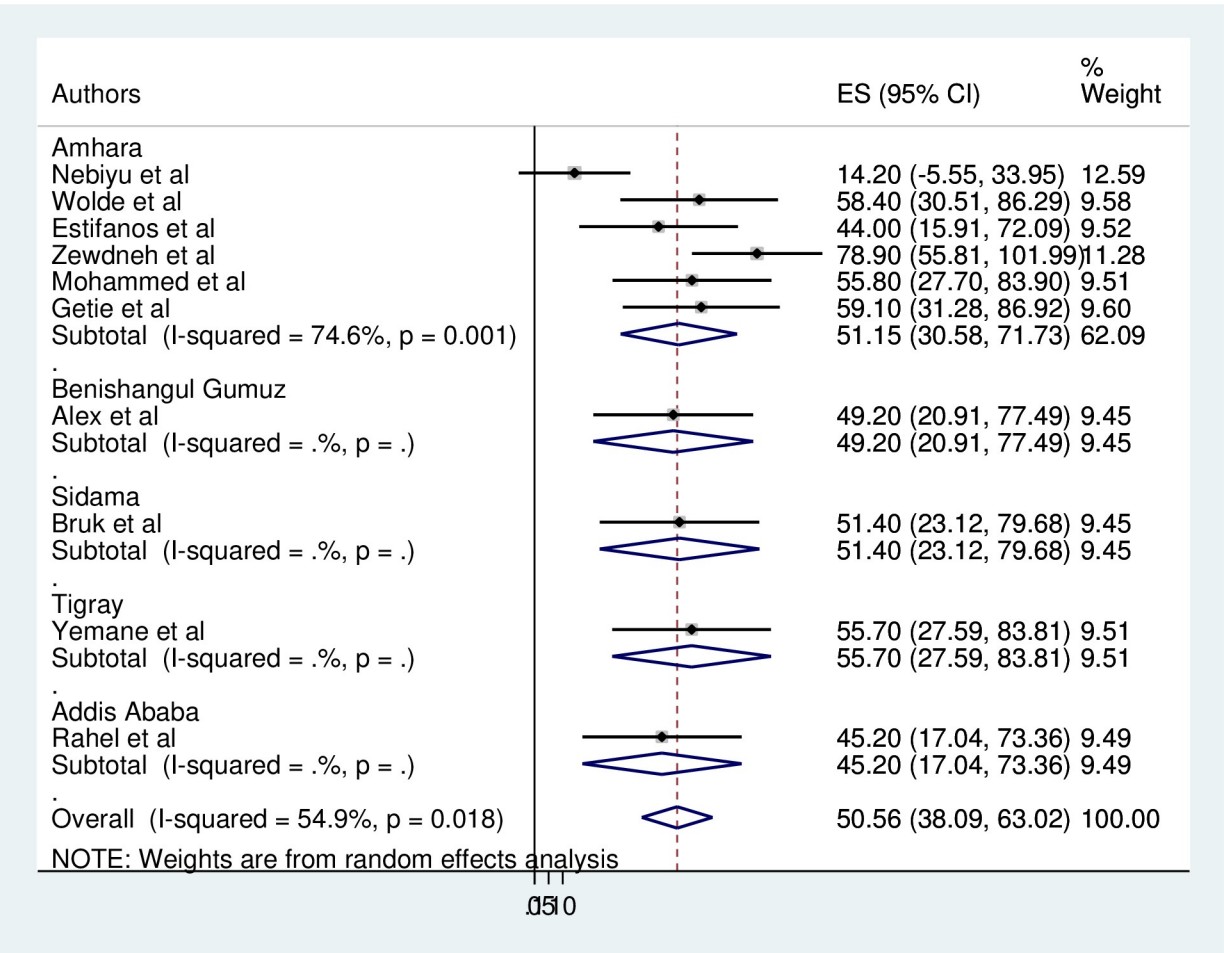

**Fig 4. Subgroup analysis on magnitude of consistent condom use among HIV/AIDS patients by their region year in Ethiopia, 2023.**

### Factors associated with a consistent condom use in Ethiopia

**Association between consistent condom use and HIV disclosure status.** To identify the association between consistent condom use and HIV disclosure status, two studies were included in the meta-analysis [1, 21]. Two of the included studies showed that disclosing the HIV status was significantly associated with consistent condom use [1, 21].

The pooled finding of the meta-analysis showed that disclosing their HIV status was significantly associated with consistent condom use. HIV patients who disclosed their status was 5.61 times have higher odds of using condom consistently as compared to HIV patients who did not disclose their status (AOR = 5.61, 95%, CI: 2.29–13.73) (Fig 7).

**Association between consistent condom use and level of education.** To identify the association between consistent condom use and educational status, five studies were included in the meta-analysis [1, 6, 10, 12, 13]. All of the included studies showed that higher level of education was significantly associated with consistent condom use [1, 6, 10, 12, 13]. However, there was no significant association between level of education and consistent condom use from their pooled findings (Fig 8).

**Association between consistent condom use and counseling on condom use.** To identify the association between consistent condom use and counseling on condom use, three

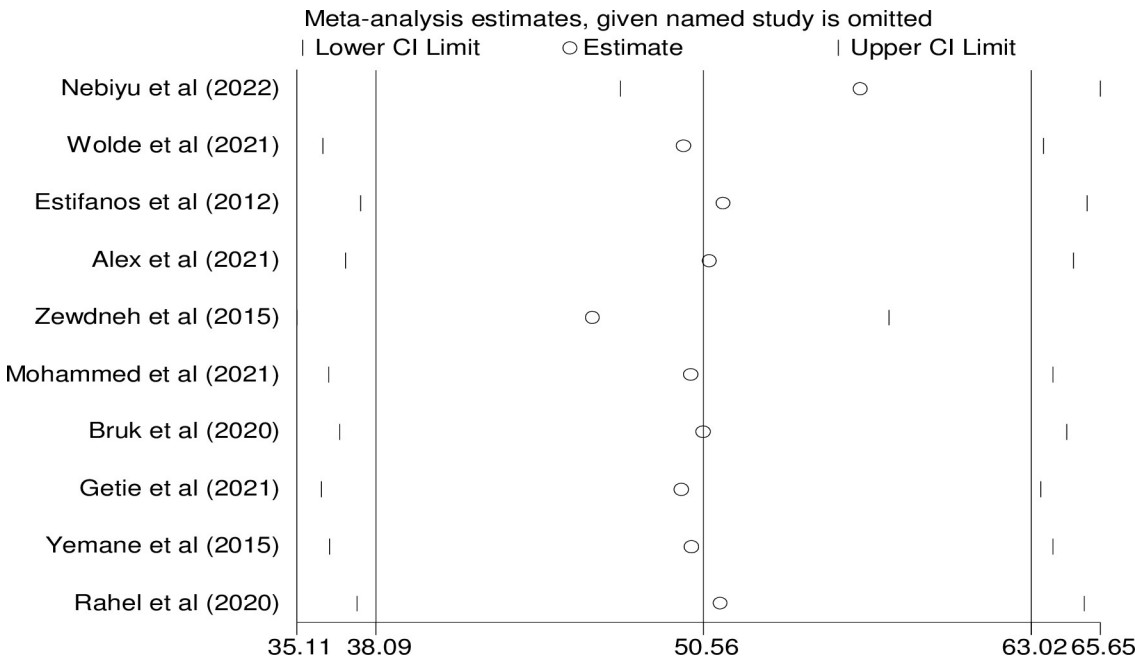

**Fig 5. Sensitivity analysis for single study influence on the overall study of consistent condom use in Ethiopia, 2023.**

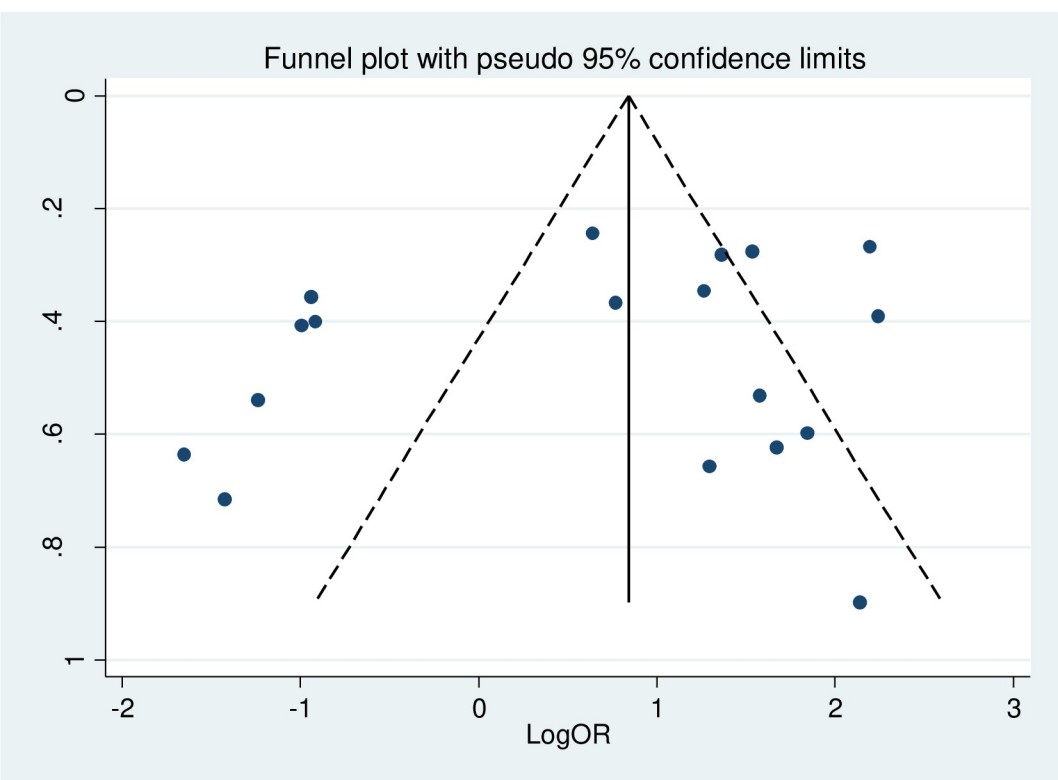

**Fig 6. Funnel plot of the included studies to test publication bias in Ethiopia, 2023.**

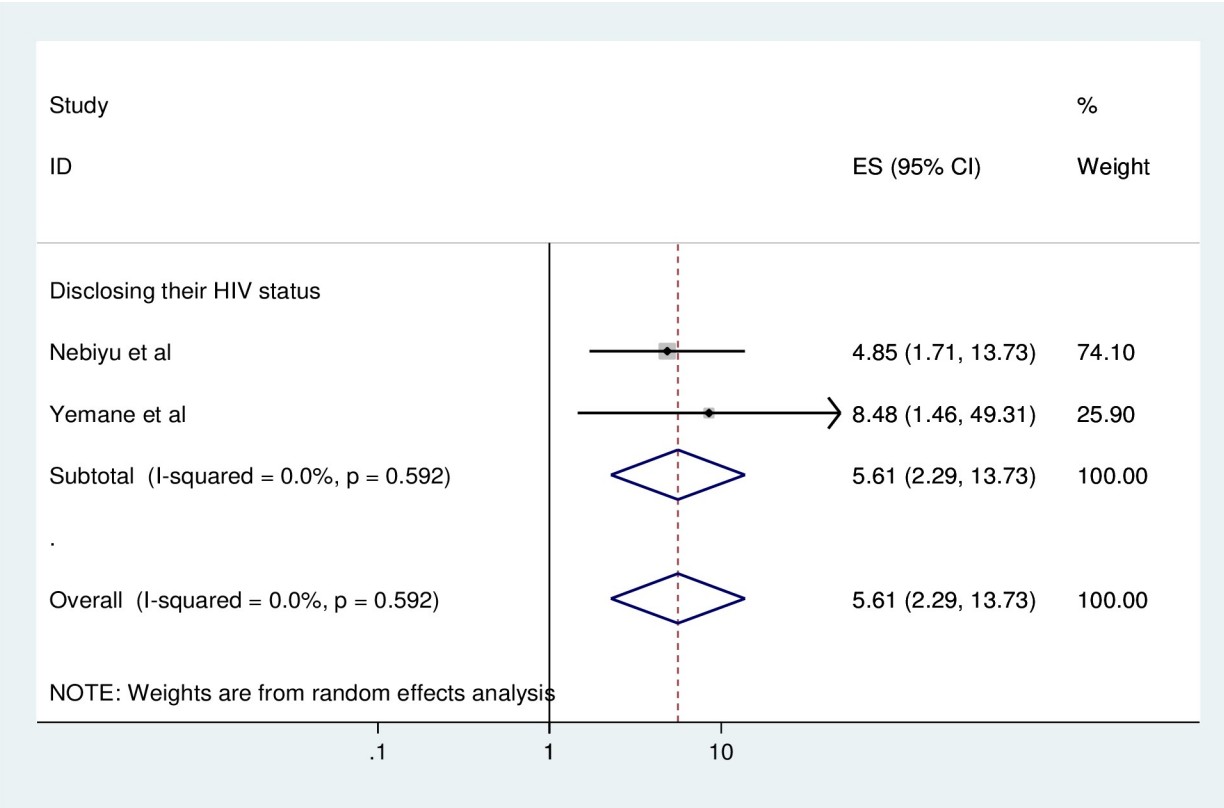

**Fig 7. Forest plot of association between consistent condom use and HIV disclosure status in Ethiopia, 2023.**

studies were included in the meta-analysis [2, 19, 20]. All of the included studies showed that counseling on condom use was significantly associated with consistent condom use [2, 19, 20]. However, there was no significant association between counseling on condom use and consistent condom use from their pooled findings (Fig 9).

**Association between consistent condom use and area of residence.** To identify the association between consistent condom use and area of residence, three studies were included in the meta-analysis [10, 6, 19]. Three of the included studies showed that urban residence was significantly associated with consistent condom use [10, 6, 19].

The pooled finding of the meta-analysis showed that urban residence was significantly associated with consistent condom use. HIV patients who live in urban area were 3.46 times have higher odds of using consistent condom use compared to rural areas (AOR = 3.46, 95%, CI: 2.24–5.35) (Fig 10).

**Association between consistent condom use and marital status.** To identify the association between consistent condom use and marital status, two studies were included in the meta-analysis [10, 24]. Two of the included studies showed that being a married was significantly associated with consistent condom use [10, 24]. The pooled finding of the meta-analysis showed that being a married was significantly associated with consistent condom use. HIV patients who were married were 67% times have lower odds of using condom consistently as compared to their counterparts (AOR = 0.33, 95%, CI: 0.18–0.61) (Fig 11).

**Association between consistent condom use and duration of ART.** To identify the association between consistent condom use and duration of ART, two studies were included in the meta-analysis [6, 21]. All of the included studies showed that longer ART duration was

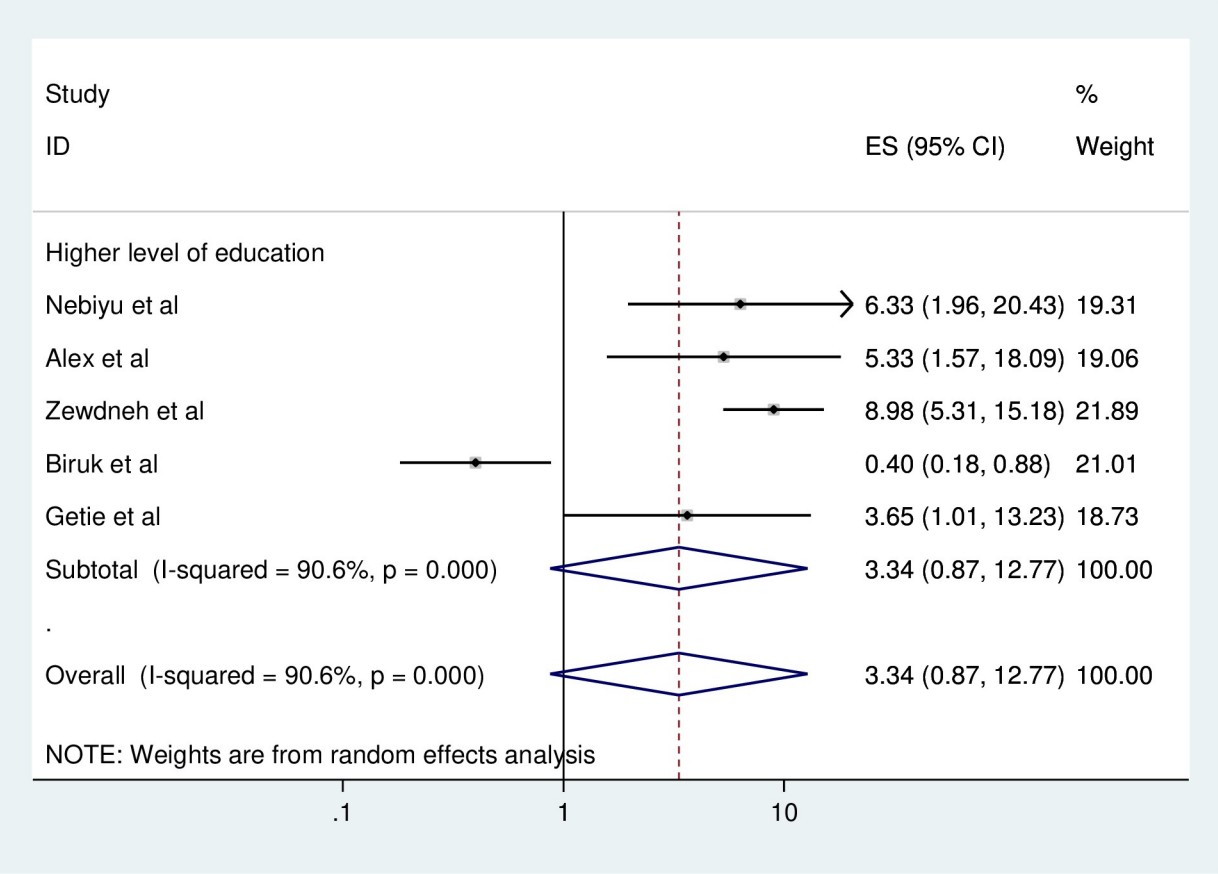

**Fig 8. Forest plot of association between consistent condom use and level of education in Ethiopia, 2023.**

significantly associated with consistent condom use [6, 21]. However, there was no significant association between ART duration and consistent condom use from their pooled findings (Fig 12).

## Discussion

To the best of our knowledge, this meta-analysis and systematic review are the first of its kind that conducted at the national level to estimates magnitude and identifies factors associated with a consistent condom use among HIV/AIDS patients in Ethiopia.

The pooled prevalence of consistent condom use among HIV/AIDS patients in Ethiopia is 50.56%. The prevalence of consistent condom use is higher than the previously conducted reviews in Ethiopia by Bewket YA et al, 2022 on prevalence of risky sexual behavior in Ethiopia that was 40% [25]. Another systematic reviews and meta-analysis done in Ethiopia by Habtamu EH et al, 2022 reported the pooled prevalence of 43.56% [26]. Similarly, the lower prevalence were reported according to the systematic review and meta-analysis conducted in Sub-Saharan Africans that was 36.16% [27]. On the contrary, the prevalence of consistent condom use was lower than the study conducted in Kenya that was 65% [9]. The lower rate of condom use in Ethiopia might be due to the sociocultural factors that may influence condom use. The other possible discrepancies could be due to the difference in the study characteristics, outcome measures, and sample size used.

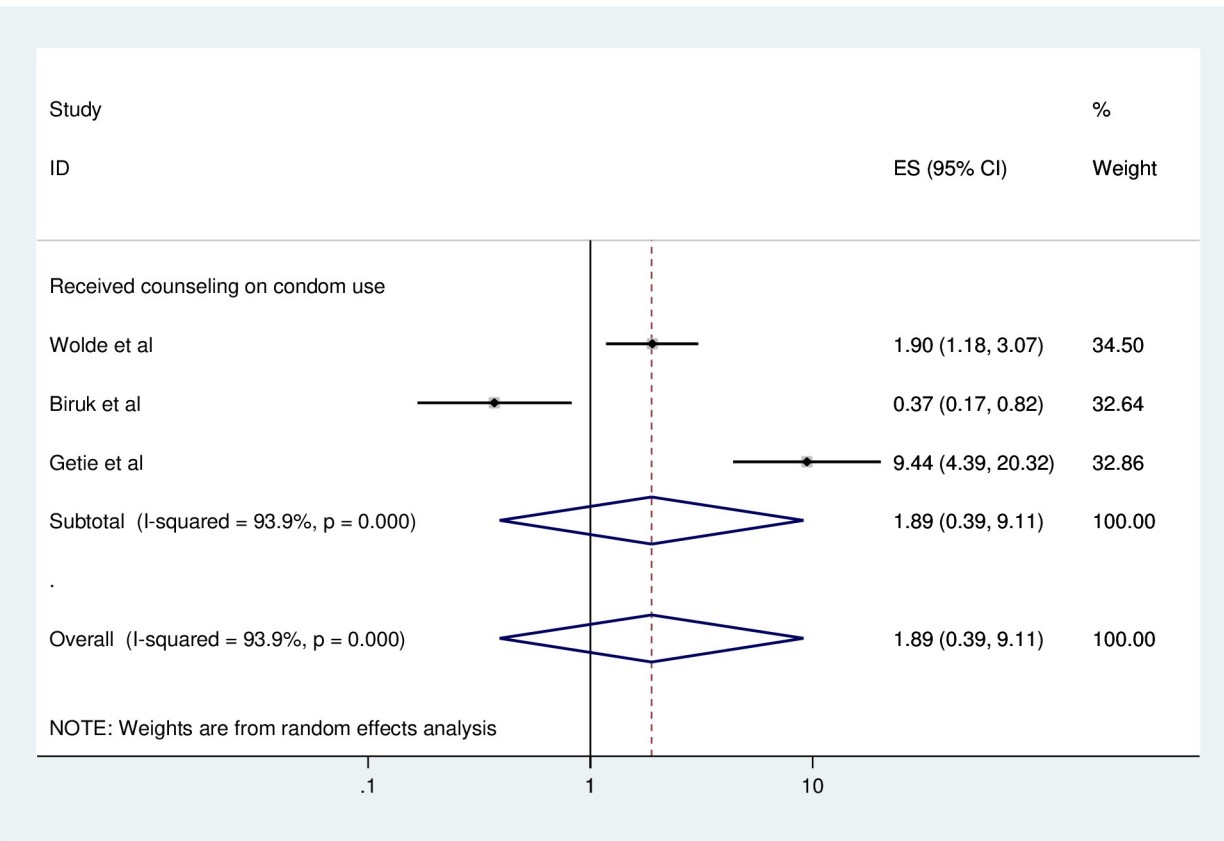

**Fig 9. Forest plot of association between consistent condom use and counseling on condom use in Ethiopia, 2023.**

The result of sub-group analysis revealed that the lower magnitude of condom utilization was found in Addis Ababa 45.2% [24] and the highest condom utilization was found in Tigray region 55.70% [21]. This might be due to low attitude and unwillingness of the peoples in the city to use condom during sexual intercourse with their partner. Therefore, the Ethiopian ministry of health should give emphasis for Addis Ababa region.

In addition, the systematic review and meta-analysis discovered variables linked to regular condom use among HIV/AIDS patients. HIV patients who were married were 67% times have lower odds of using condom consistently as compared to their counterparts. On the other hand, compared to married couples, unmarried couples were less likely to consistently use condoms, according to research done in Portugal [28] and Zambia [15]. The results of our investigation were in line with a study conducted in Nigeria, where respondents who were bereaved or single were more likely to use condoms [29]. This is because single or widowed individuals are more prone to get STIs or 'other' subtypes of the virus since they are less likely to engage in committed relationships in general. Additionally, it's conceivable that the medical personnel at the clinic has concentrated its condom

HIV patients who disclosed their status was 5.61 times have higher odds of using condom consistently as compared to HIV patients who did not disclose their status. The systematic review and meta-analysis in Sub-Saharan Africans also revealed that non-disclosure of their HIV status were associated with inconsistent condom use [27]. Similarly, a systematic reviews and meta-analysis done in Ethiopia by Habtamu EH et al, 2022 showed that discussing about safe sex with sexual partner was a predictors of risky-sexual behavior [26]. This could occur as

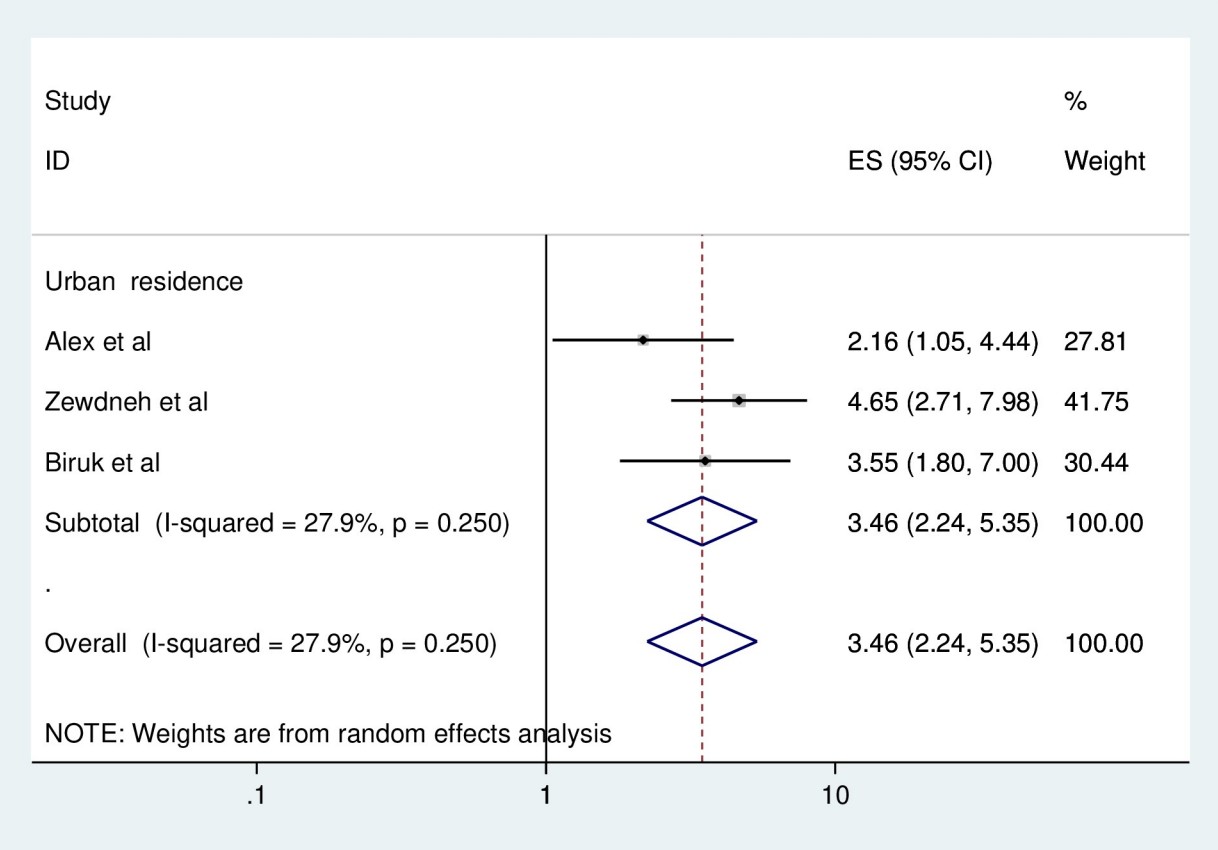

**Fig 10. Forest plot of association between consistent condom use and urban residence in Ethiopia, 2023.**

a result of couples learning more about how to avoid HIV transfer, infection, and reinfection in order to protect their unborn child from diseases by discussing the advantages of condom use with sexual partners.

The place of residence was predictors of consistent condom use. HIV patients who live in urban area were 3.46 times have higher odds of using consistent condom use compared to rural areas. The investigation carried out in Cameroon [30] did not yield a similar outcome. The results, however, were in line with those of Nigeria [31] and Thailand [32]. Patients who reside in urban areas might have learned about condom use and had access to medical facilities where they could receive appropriate counseling, and the medical facilities might raise public awareness of the significance of consistent condom use, which is targeted at high-risk individuals who reside in and around the urban.

The current research has significant clinical ramifications for stopping the spread of STIs and novel strains of HIV. STI and new strain of HIV prevention techniques and initiatives should concentrate on married couples, those who live in rural regions, and those who did not reveal their sero-status.

In order to prevent STIs, PLHIV with several partners must always be encouraged to use condoms, with an emphasis on secondary pairings. These demonstrate the need for HIV programs that promote information sharing between sexual partners and enhance condom-specific and transmission prevention education for people living with HIV [9]. The finding could assist policymakers in reducing the risk of infection and re-infection that occur from

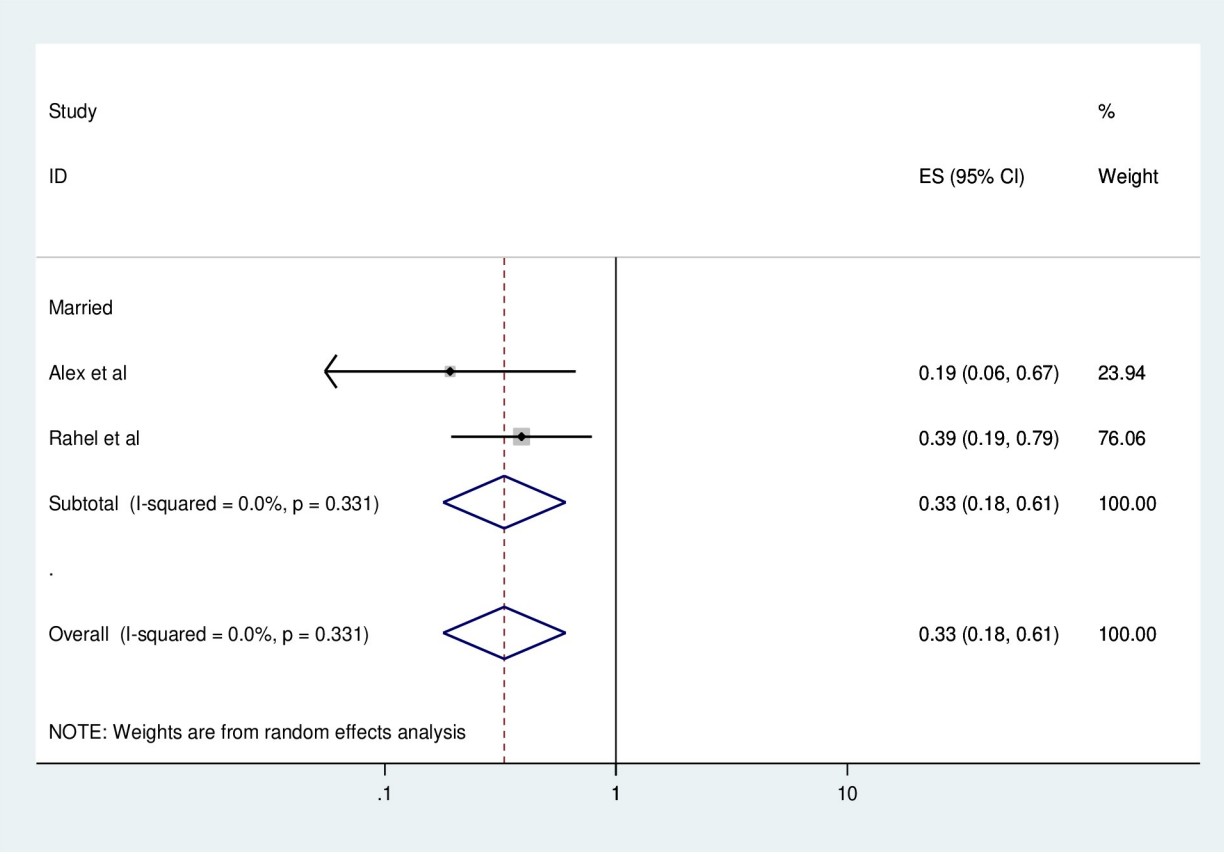

**Fig 11. Forest plot of association between consistent condom use and marital status in Ethiopia, 2023.**

inconsistent condom use. The educational interventions through different media could help increase the use of condoms in Ethiopia.

The STI prevention campaign, which primarily focuses on raising awareness about condom use, should be implemented by the government and decision-makers. Preventive programs should also include a variety of interactive health education through different media for optimal effectiveness [33]. This study also advises that the ministry of health should support and encourage consistent condom use among HIV/AIDS patients.

## Limitation of the study

As the limitation, the sample size of the included studies was small, the heterogeneity was found among included studies and risk of bias was not reported. In addition, all of the studies included in this review were cross-sectional study design; as a result, the causal effect relationship was could not be identified. Finally, the primary study was not conducted in whole region of Ethiopia to include in our systematic and meta-analysis.

## Conclusion

Half of the patients with HIV/AIDS used condoms consistently according to our systematic review and meta-analysis study. According to this study, condom use among HIV/AIDS patients who live in metropolitan areas, are disclosing their status, and are married are all predicted. Thus, married couples and those living in rural regions should receive health education

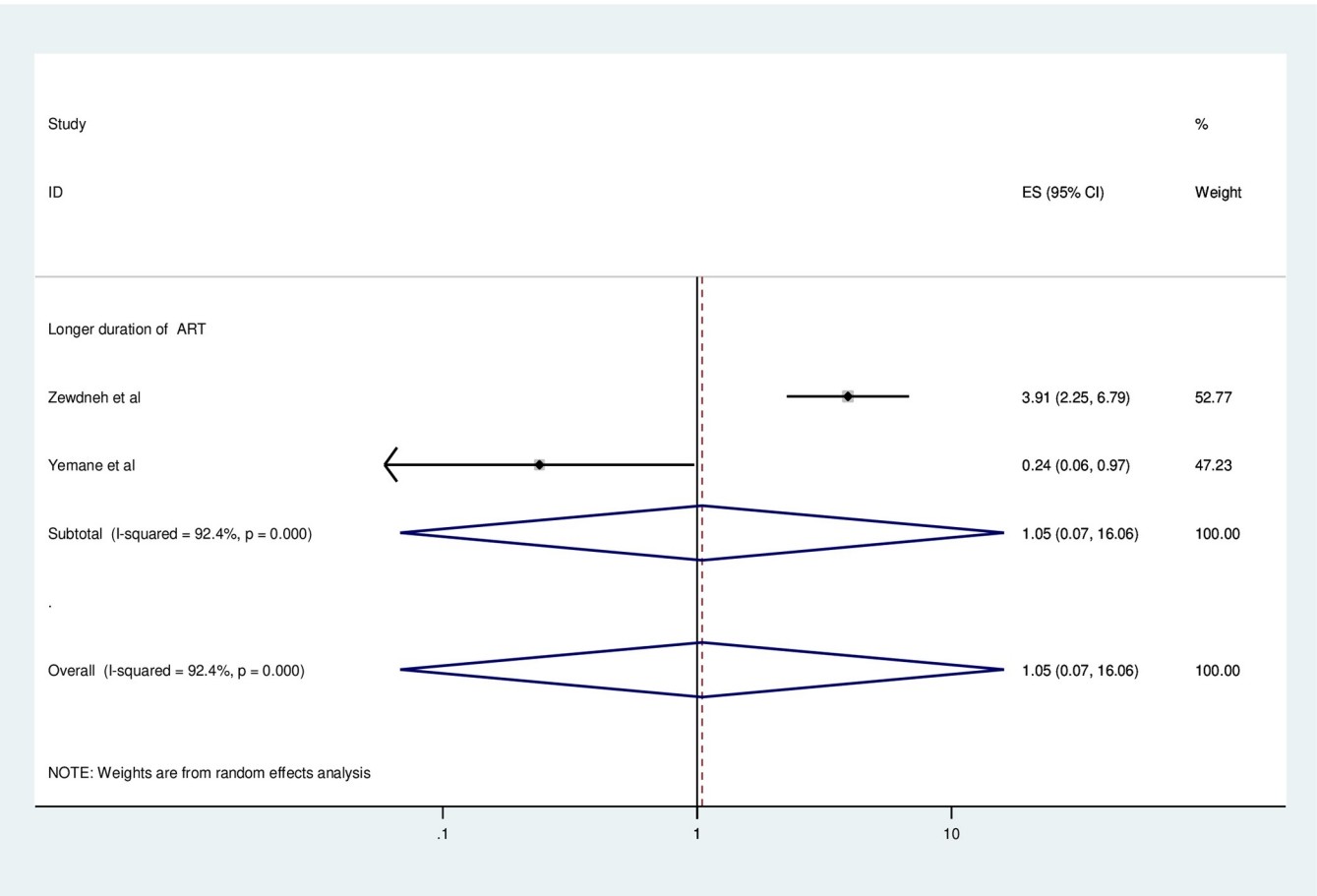

**Fig 12. Forest plot of association between consistent condom use and ART duration in Ethiopia, 2023.**

and counseling services about using condoms consistently. Furthermore, it's possible that clinic staff members focused more of their condom and preventative teachings on married PLHIV couples than they did on widowed and single couples. Besides this, disclosing their HIV status to their sexual partners and about the need for consistent condom use would be important in consistent condom utilization. However, there was no significant association between consistent condom use and counseling on condom use, duration of ART, and level of education. The effect of a single study on the overall outcome was not found from the sensitivity analysis report. The results of sub-group analysis revealed a heterogeneity in relation to their geographical region. The lowest prevalence of consistent condom use was reported in Addis Ababa region. Therefore, special attention should be given for peoples residing in this region.

## Supporting information

**S1 Checklist. PRISMA-2020 (Preferred Reporting Items for Systematic Reviews and Meta-Analysis-2020) checklist.**
(DOCX)

**S1 File. Search strategies.**
(DOCX)

## Acknowledgments

We would like to thank all authors of the studies included in this systematic review and meta-analysis.

## Author Contributions

**Conceptualization:** Firomsa Bekele, Teshome Debushe.

**Data curation:** Lalise Tafese.

**Formal analysis:** Firomsa Bekele.

**Investigation:** Firomsa Bekele, Lalise Tafese.

**Methodology:** Firomsa Bekele, Lalise Tafese, Teshome Debushe.

**Writing – original draft:** Teshome Debushe.

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
