## [Decision Letter · Decision Letter 0]

24 Jul 2023

PONE-D-23-16392Magnitude of consistent condom use and associated factors among peoples living with HIV/AIDS in Ethiopia: Implication for reducing infections and re-infection. A systematic review and meta-analysisPLOS ONE

Dear Dr. Bekele,

Thank you for submitting your manuscript to PLOS ONE. After careful consideration, we feel that it has merit but does not fully meet PLOS ONE’s publication criteria as it currently stands. Therefore, we invite you to submit a revised version of the manuscript that addresses the points raised during the review process.

We look forward to receiving your revised manuscript.

Kind regards,

Chuanyi Ning, Ph.D.

Academic Editor

PLOS ONE

Journal Requirements:

https://link.springer.com/article/10.1186/s12889-019-8133-y

https://www.researchgate.net/publication/354994362_Prevalence_of_Consistent_Condom_Use_and_Associated_Factors_among_Serodiscordant_Couples_in_Ethiopia_2020_A_Mixed-Method_Study

In your revision ensure you cite all your sources (including your own works), and quote or rephrase any duplicated text outside the methods section. Further consideration is dependent on these concerns being addressed.

Reviewers' comments:

Reviewer's Responses to Questions

**Comments to the Author**

1. Is the manuscript technically sound, and do the data support the conclusions?

Reviewer #1: Partly

Reviewer #2: Partly

2. Has the statistical analysis been performed appropriately and rigorously? 

Reviewer #1: No

Reviewer #2: Yes

3. Have the authors made all data underlying the findings in their manuscript fully available?

Reviewer #1: Yes

Reviewer #2: Yes

4. Is the manuscript presented in an intelligible fashion and written in standard English?

Reviewer #1: No

Reviewer #2: No

5. Review Comments to the Author

Reviewer #1: - Thank you for requesting my opinion on this work. The authors conducted a systematic review and meta-analysis to determine the magnitude of consistent condom use and associated factors among people living with HIV/AIDS in Ethiopia and their implications for reducing infections and re-infection. However, several distinct systematic reviews and meta-analyses regarding risky sexual behaviour among HIV/AIDS patients have been conducted in Ethiopia. These studies used a similar operational definition and cited inconsistent condom use as a risky sexual practise. Additionally, I was unable to gain any new information or insights from this study, and the authors were unable to present concrete evidence that there was a literature gap or any other factors that made the current research different from previously published publications.

- The authors used the terms "adherence to self-care" and "consistently used condoms" interchangeably; I'm not sure these terms are similar, and I believe they have different meanings in this context. Consistent condom use may be one self-care practise that people with HIV/AIDS engage in, and self-care is a holistic approach. I advise the authors to employ one of them consistently throughout the whole paper.

- What are the gaps? I've noticed that the introduction is very poor and insufficient, and there is no clear, updated context about the current situation of HIV/AIDS in Ethiopia. Moreover, the report fails to explain how it differs from earlier comparable systematic reviews and meta-analyses. Additionally, most of the references are incorrectly cited. For instance, the authors used reference number 3 to illustrate the prevalence of HIV/AIDS in Ethiopia, but the cited study was unable to provide this data; as a result, the correct data for this information can be found in this article/Ababa A. Ethiopia. Abstract available from: https://wfpha.confex.com/wfpha/2012/webprogram/Paper10587.html. 2013.

- The author said that the review procedure was registered on PROSPERO CRD430396 in the method section. However, the registration digits were missing; therefore, if the review was registered, I advise the author to supply the whole registration number.

- Search strategies: It would be good practice to include the search strategies and the number of hits for each line as an appendix or supplementary material.

- If possible, could you elaborate on the inclusion and exclusion criteria under the heading "Eligibility criteria" for the review? How about the primary study's choice of study design?

- The Joanna Briggs Institute Meta-analysis of Statistics Assessment and Review Instrument (JBIMAStARI) was used by the authors to evaluate the quality of their work; however, they were unable to describe how to utilise it. no explanation of the criteria used for assessing quality. When do you say good quality article?" Could you please describe how to use the tool? How many reviewers, for instance, completed the quality assessment? Additionally, I didn't see any reports about quality evaluation in the results section.

- In this final method part subheading, authors wrote about the risk of bias. I'm not sure what the authors are saying, it sounds like personal opinion. Please provide concise support and write it better in the analysis portion with the proper reference.

- The data analysis and synthesis sections were extremely short, and all the plans utilised were not cited. In addition, the authors didn't specify which model to apply (fixed effect or random); what about the pan-sub-group and sensitivity analyses? What is the evidence for claiming heterogeneity at a p-value of 0.05?

- in the result, according to the criteria in the method section, your study's heterogeneity was moderate. What does it stand for? Describe it in detail, please. Describe the probable effects of this moderate degree of heterogeneity on the pooled prevalence estimate and discuss any possible causes of this moderate level of heterogeneity.

- In discussion section, how did the authors ensure that, to the best of their knowledge, this meta-analysis and systematic review are the first of their kind to be conducted at the national level in Ethiopia to estimate the magnitude and identify the factors associated with consistent condom use among HIV/AIDS patients, despite the fact that many other studies of a similar nature have been carried out in Ethiopia?

- The discussion portion is too short and lacks comparison of the details with the existing evidence. The authors failed to highlight the clinical implications, and I was unable to locate the implications of this study for stakeholders and policymakers. They also made no recommendations for these groups of people.

- Finally, I advise the author to thoroughly review every area of the work and seek the advice of professionals in grammar and language.

Reviewer #2: This paper explores extent of consistent condom use and factors associated with consistent condom use among people living with HIV in Ethiopia. Although the topic is important, there are several issues in presentation of the data. There has been inconsistent use of the language, and language appropriateness. Overall, Although methodologically correct, It needs to be thoroughly reviewed by an English speaker. Below are few examples of the areas need to be revised, however, there is a lot more than what I singled out.

Comments

Abstract:

Results section there is a sentence “ the pooled estimate of self-care (What is self-care? It has never been defined earlier. I expected pooled prevalence of consistent condom use instead of self-care)

Conclusion: This study shown that (it should be showed)

“telling their sexual partners about their” Disclosing HIV status is better than telling their partners…..

Background:

…..is dependent on the consistent and proper use of condoms(2, 4-6). However, people who started (There is no connection with the use of word however here.)

Sub-Saharan Africa, especially Ethiopia, has a very high incidence of the problem (2). (Better state it explicitly instead of referring it as a problem

Despite a variable reports of magnitude and…… (Grammatically incorrect)

Methods

The objective of the review was to conclude the magnitude and associated factors of consistent

condom use among peoples living with HIV/AIDS in Ethiopia (This needs revision such as ……..magnitude of consistent condom use and factors associated with consistent condom use……..)

MESH term for the database is ( should be MESH term for the database were…..)

Discussion

The rate of adherence ….. (Adherence to what? better use same word consistent cdm use.)

patients who reported their status (disclosed? reported to who?)

This result was at odds with the research on Cameroon (I don’t understand what is meant here)

6. PLOS authors have the option to publish the peer review history of their article (what does this mean?). If published, this will include your full peer review and any attached files.

Reviewer #1: No

Reviewer #2: No

---

## [Author Response · Author response to Decision Letter 0]

29 Aug 2023

Chuanyi Ning, Ph.D.

Academic Editor of PLOS ONE

Dear Editor of the Manuscript PONE-D-23-16392 Magnitude of consistent condom use and associated factors among peoples living with HIV/AIDS in Ethiopia: Implication for reducing infections and re-infection. A systematic review and meta-analysis" submitted to PLOS ONE. Thanks for your time and consideration in editing and reviewing the manuscript. We have carefully read your comments and corrected inline of your comments and suggestions. All comments raised were edited and incorporated in the revised manuscript. 

Here are the responses and elaborations for the comments from the editor and reviewer!

REVIEWERS COMMENT

Reviewer 1: 

Reviewer comment: several distinct systematic reviews and meta-analyses regarding risky sexual behaviour among HIV/AIDS patients have been conducted in Ethiopia. These studies used a similar operational definition and cited inconsistent condom use as a risky sexual practice. Additionally, I was unable to gain any new information or insights from this study, and the authors were unable to present concrete evidence that there was a literature gap or any other factors that made the current research different from previously published publications.

Author response: Risky sexual behaviour is the general term but inconsistent condom use is too specific term. Therefore, those two studies are quite different that makes our study unique from risky sexual behaviour. Many authors defined risky sexual behaviour as having multiple sexual partners, early initiation of sex, inconsistent condom use, sex with commercial sex workers, and unprotected sex with a same-sex partner, especially when it involves male partners 

Reviewer comment: The authors used the terms "adherence to self-care" and "consistently used condoms" interchangeably; I'm not sure these terms are similar, and I believe they have different meanings in this context. Consistent condom use may be one self-care practise that people with HIV/AIDS engage in, and self-care is a holistic approach. I advise the authors to employ one of them consistently throughout the whole paper.

Author response: We have used consistent condom use instead of adherence to self-care throughout the paper 

Reviewer comment: What are the gaps? I've noticed that the introduction is very poor and insufficient, and there is no clear, updated context about the current situation of HIV/AIDS in Ethiopia. Moreover, the report fails to explain how it differs from earlier comparable systematic reviews and meta-analyses. Additionally, most of the references are incorrectly cited. For instance, the authors used reference number 3 to illustrate the prevalence of HIV/AIDS in Ethiopia, but the cited study was unable to provide this data; as a result, the correct data for this information can be found in this article/Ababa A. Ethiopia. Abstract available from: https://wfpha.confex.com/wfpha/2012/webprogram/Paper10587.html. 2013.

Author response: The revised manuscript contains of the study gaps, and sufficient background information. The updated data on the current situation of HIV/AIDS was added.The difference between the pervious study and our study was stated in introduction part as ”In Ethiopia, there was no systematic review and meta-analysis was conducted regarding the magnitude and predictors of consistent condom use, and the previous comparable systematic reviews and meta-analyses done was on risky sexual behavior that includes having multiple sexual partners, early initiation of sex, sex with commercial sex workers, and unprotected sex with a same-sex partner, which is different from consistent condom use. Our study only reports on condom utilization patterns. Therefore, the purpose of this paper is to summarize the recent findings on magnitude and factors related to consistent condom use in order to provide an appropriate intervention”.

The reference no 3 was replaced by another updated references

Reviewer comment: The author said that the review procedure was registered on PROSPERO CRD430396 in the method section. However, the registration digits were missing; therefore, if the review was registered, I advise the author to supply the whole registration number.

Author response: The review was registered and given a registration no of PROSPERO CRD42023430396

Reviewer comment: Search strategies: It would be good practice to include the search strategies and the number of hits for each line as an appendix or supplementary material.

Author response: The Search strategies was supplied as supplementary material in revised manuscript

Reviewer comment: If possible, could you elaborate on the inclusion and exclusion criteria under the heading "Eligibility criteria" for the review? How about the primary study's choice of study design?

Author response: We have clearly explain it in revised manuscript

Reviewer comment: The Joanna Briggs Institute Meta-analysis of Statistics Assessment and Review Instrument (JBIMAStARI) was used by the authors to evaluate the quality of their work; however, they were unable to describe how to utilise it. no explanation of the criteria used for assessing quality. When do you say good quality article?" Could you please describe how to use the tool? How many reviewers, for instance, completed the quality assessment? Additionally, I didn't see any reports about quality evaluation in the results section.

Author response: We have corrected in quality assessment section of revised manuscript as” The criteria used for assessing the quality of included studies were as follows; Were the criteria for inclusion in the sample clearly defined?, Were the study subjects and the setting described in detail?, Was the exposure measured in a valid and reliable way?, Were objective, standard criteria used for measurement of the condition?, Were confounding factors identified?, Were strategies to deal with confounding factors stated?, Were the outcomes measured in a valid and reliable way?, and Was appropriate statistical analysis used?. The quality assessment were done by two authors (FB and GB) using a quality assessment checklist. Discrepancies between two independent reviewers were resolved by involving third reviewers (TD) after discussion for possible consensus. Accordingly, the articles was classified as high quality if the score is >80%, moderate if (65-80%), and low if <65%. The reports about quality evaluation was described in table 2 of revised manuscript under results section.

Reviewer comment: In this final method part subheading, authors wrote about the risk of bias. I'm not sure what the authors are saying, it sounds like personal opinion. Please provide concise support and write it better in the analysis portion with the proper reference.

Author response: The risk of bias section is omitted and stated as limitation in revised manuscript

Reviewer comment: The data analysis and synthesis sections were extremely short, and all the plans utilised were not cited. In addition, the authors didn't specify which model to apply (fixed effect or random); what about the pan-sub-group and sensitivity analyses? What is the evidence for claiming heterogeneity at a p-value of 0.05?

Author response: The data analysis section was revised to include all the plan and model used, sub-group analysis and sensitivity analysis

Reviewer comment: in the result, according to the criteria in the method section, your study's heterogeneity was moderate. What does it stand for? Describe it in detail, please. Describe the probable effects of this moderate degree of heterogeneity on the pooled prevalence estimate and discuss any possible causes of this moderate level of heterogeneity.

Author response: We have corrected in revised manuscript as “The moderate heterogeneity showed the presence of moderate variations across studies rather than by chance. This heterogeneity can alter the pooled magnitude estimate of consistent condom use. Therefore, subgroup analysis and sensitivity analysis was carried out. The region was the possible causes of this moderate heterogeneity. The possible reasons for heterogeneity, other than clinical differences, might include methodological issues such as problems with sampling technique”.

Reviewer comment: In discussion section, how did the authors ensure that, to the best of their knowledge, this meta-analysis and systematic review are the first of their kind to be conducted at the national level in Ethiopia to estimate the magnitude and identify the factors associated with consistent condom use among HIV/AIDS patients, despite the fact that many other studies of a similar nature have been carried out in Ethiopia?

Author response: There was no systematic review and meta-analysis study conducted on magnitude and associated factors with consistent condom use among HIV/AIDS patients in Ethiopia. The already conducted study was on risky sexual behavior which is different from our study

Reviewer comment: The discussion portion is too short and lacks comparison of the details with the existing evidence. The authors failed to highlight the clinical implications, and I was unable to locate the implications of this study for stakeholders and policymakers. They also made no recommendations for these groups of people.

Author response: The discussion was modified and extensive information was added. Besides, the clinical implications and recommendation were added.

Reviewer comment: Finally, I advise the author to thoroughly review every area of the work and seek the advice of professionals in grammar and language.

Author response: The whole document was revised and the grammar was edited by one fluent English speaker. 

Reviewer 2:

Reviewer comment: There has been inconsistent use of the language, and language appropriateness. Overall, Although methodologically correct, It needs to be thoroughly reviewed by an English speaker.

Author response: The whole document was revised and the grammar was edited by one fluent English speaker. 

Reviewer comment: Results section there is a sentence “ the pooled estimate of self-care (What is self-care? It has never been defined earlier. I expected pooled prevalence of consistent condom use instead of self-care)

Author response: The word self-care was replaced by consistent condom use

Reviewer comment: Conclusion: This study shown that (it should be showed)

Author response: It was corrected as showed

Reviewer comment: “telling their sexual partners about their” Disclosing HIV status is better than telling their partners…..

Author response: We have corrected as Disclosing HIV status.

Reviewer comment: .is dependent on the consistent and proper use of condoms(2, 4-6). However, people who started (There is no connection with the use of word however here.)

Author response: The sentence was re-written in revised manuscript

Reviewer comment: Sub-Saharan Africa, especially Ethiopia, has a very high incidence of the problem (2). (Better state it explicitly instead of referring it as a problem

Author response: We have mentioned the problem in the revised manuscript

Reviewer comment: Despite a variable reports of magnitude and…… (Grammatically incorrect)

Author response: We have corrected its grammar in the revised manuscript.

Reviewer comment: The objective of the review was to conclude the magnitude and associated factors of consistent

condom use among peoples living with HIV/AIDS in Ethiopia (This needs revision such as ……..magnitude of consistent condom use and factors associated with consistent condom use……..)

Author response: We have revised it as per your comment

Reviewer comment: MESH term for the database is ( should be MESH term for the database were…..)

Author response: Is was replaced by were 

Reviewer comment: The rate of adherence ….. (Adherence to what? better use same word consistent cdm use.)

Author response: We have replaced adherence by consistent condom use

Reviewer comment: patients who reported their status (disclosed? reported to who?)

Author response: We have corrected as patients who disclosed their status to their partner in revised manuscript

Reviewer comment: This result was at odds with the research on Cameroon (I don’t understand what is meant here)

Author response: We mean different. Therefore, we have replaced odds by different in revised manuscript

Thanks for your time and consideration,

 Regards!

---

## [Decision Letter · Decision Letter 1]

23 Jan 2024

PONE-D-23-16392R1Magnitude of consistent condom use and associated factors among peoples living with HIV/AIDS in Ethiopia: Implication for reducing infections and re-infection. A systematic review and meta-analysisPLOS ONE

Dear Dr. Negera,

Thank you for submitting your manuscript to PLOS ONE. After careful consideration, we feel that it has merit but does not fully meet PLOS ONE’s publication criteria as it currently stands. Therefore, we invite you to submit a revised version of the manuscript that addresses the points raised during the review process.

We look forward to receiving your revised manuscript.

Kind regards,

Nagasa Dida, MPH

Academic Editor

PLOS ONE

Journal Requirements:

Reviewers' comments:

Reviewer's Responses to Questions

**Comments to the Author**

1. If the authors have adequately addressed your comments raised in a previous round of review and you feel that this manuscript is now acceptable for publication, you may indicate that here to bypass the “Comments to the Author” section, enter your conflict of interest statement in the “Confidential to Editor” section, and submit your "Accept" recommendation.

Reviewer #1: (No Response)

Reviewer #2: (No Response)

Reviewer #3: All comments have been addressed

2. Is the manuscript technically sound, and do the data support the conclusions?

Reviewer #1: Partly

Reviewer #2: Yes

Reviewer #3: Yes

3. Has the statistical analysis been performed appropriately and rigorously? 

Reviewer #1: Yes

Reviewer #2: Yes

Reviewer #3: Yes

4. Have the authors made all data underlying the findings in their manuscript fully available?

Reviewer #1: Yes

Reviewer #2: Yes

Reviewer #3: Yes

5. Is the manuscript presented in an intelligible fashion and written in standard English?

Reviewer #1: Yes

Reviewer #2: No

Reviewer #3: Yes

6. Review Comments to the Author

Reviewer #1: I appreciate the editors asking me to assess the article once more; the authors did a good job of improving it. Before accepting the article for publication, I offered a few comments and suggestions that needed to be addressed.

- The authors noted that there was conflicting evidence available in Ethiopia about the prevalence and correlates of frequent condom usage in the abstract portion of the paper. Keeping this in mind, how do the authors address these problems? How do they generate fresh evidence? What were the conflicting issues that appeared in earlier literature?

- The abstract's method section has to be updated in light of the objectives: How was consistent condom usage measured and reported? How were the factors associated with condom use described and their statistical significance determined? How about sub-group analysis carried out by region or any other criteria, too? How was the objective publication bias determined?

- The numbers on the number of people living with HIV/AIDS needed to be reported along with the year, as stated in the first and second paragraphs of the introduction. At what time did this report occur?

- In the introduction section, the authors noted that different countries use condoms at varying rates. To get a better general picture of condom usage among HIV/AIDS patients, I asked the authors to describe the prevalence of condom use among people living with the disease elsewhere in the world, with a focus on sub-Saharan Africa and Ethiopia.

-be more appropriate to cite the prior comparable reviews and analyses that have concentrated on risky sexual activity, which is different from your work on regular condom use? Could you offer any alternative suggestions to the ones found in the earlier studies? On the other hand, tackling risky sexual behavior entails encouraging frequent condom usage, right? Could there be different results? I believe that all of your research and earlier studies focused on preventing HIV/AIDS and re-infection.

- on objectives: The objective of the review was to conclude the magnitude and associated factors of consistent condom use among people living with HIV/AIDS in Ethiopia. It is better to say to detrmine instead of to conclude.

- on search strategies: It would be best practice to include the search methods and the quantity of results for each database as an appendix or supplemental material. However, the authors either didn't do that or we don't have access to the supplemental table outlining the search methods and findings for each database.

- Why a publication year-based subgroup analysis? Does it affect the study in any way?

- The discussion part still requires improvement and a more thorough comparison with past similar reviews and meta-analyses conducted worldwide. The comparison between this pooled prevalence and other studies' point prevalence was the authors' sole focus. When comparing the pooled data with point prevalence, is it good?

- On the other hand, the authors calculated the pooled odds ratio of factors affecting consistently using condoms in Ethiopia. then continue comparing the findings to those of another study done in Ethiopia (Dessie)? What caused it? What's the point? They neglected to include this study in their evaluation.

- in relation to the study's implications. Pharaphrasing and citations are necessary.

- The authors needed to describe the sub-group analysis by region in the discussion section for better clarity, policy implications, and to identify the region of requirements (regional variance). can explain the differences as well as make recommendations for areas with low prevalence?

- In the limitation section? Does your study represent the whole region of Ethiopia? Are all regions of Ethiopia represented?

- The conclusion is far too short. Try to address all of the key findings and make workable suggestions.

Good luck.

Reviewer #2: Title; Change peoples to people

Abstract: More than one million peoples…. (change peoples to people)

….among HIV patients…., change it to people living with HIV or people with HIV

We have used four databases such as PubMed, when you say such as is like you are giving examples of the databases but not exact databases you searched information. Revise this sentence because those databases are the one you sought information

Main text

Methods: …Accordingly, the articles was (should be article was or articles were)

Results …majority of the participant was female in all articles ( should be majority of participants were..)

….who were disclosed their status was 5.61 time … (should be who disclosed their status…)

…patients who were live in urban area was 3.46 time (should be … who live in urban area were 3.46 times … or who are living in urban area were …..)

…patients who were married was 67% times ( should be patients who were married were 67% times…)

In addition, the measure of effect used was odd ratio, however, the interpretation is like risk ratio.

It should be something like people living in urban areas have higher odds or reporting consistent condom use compared to rural areas. Not saying people living in urban areas are more likely reporting consistent condom use compared to rural areas.

Reviewer #3: Thank you for addressing the points highlighted in the previous review. The manuscript reads well and is technically sound.

7. PLOS authors have the option to publish the peer review history of their article (what does this mean?). If published, this will include your full peer review and any attached files.

Reviewer #1: No

Reviewer #2: No

Reviewer #3: No

---

## [Author Response · Author response to Decision Letter 1]

12 Feb 2024

AUTHOR RESPONSE LETTER

 Nagasa Dida, MPH

Academic Editor of PLOS ONE

Dear Editor of the Manuscript PONE-D-23-16392R1 entitled “Magnitude of consistent condom use and associated factors among peoples living with HIV/AIDS in Ethiopia: Implication for reducing infections and re-infection. A systematic review and meta-analysis" submitted to PLOS ONE. Thanks for your time and consideration in editing and reviewing the manuscript. We have carefully read your comments and corrected inline of your comments and suggestions. All comments raised were edited and incorporated in the revised manuscript. 

Here are the responses and elaborations for the comments from the editor and reviewer!

REVIEWERS COMMENT

Reviewer 1: 

Reviewer comment: The authors noted that there was conflicting evidence available in Ethiopia about the prevalence and correlates of frequent condom usage in the abstract portion of the paper. Keeping this in mind, how do the authors address these problems? How do they generate fresh evidence? What were the conflicting issues that appeared in earlier literature?

Author response: There was no conflicts of ideas but we mean to explain in revised manuscript as there is inconsistent reports of the prevalence and correlates of frequent condom use in Ethiopia. 

Reviewer comment: The abstract's method section has to be updated in light of the objectives: How was consistent condom usage measured and reported? How were the factors associated with condom use described and their statistical significance determined? How about sub-group analysis carried out by region or any other criteria, too? How was the objective publication bias determined?

Author response: We have updated abstract's method section in line with our objectives in revised manuscript

Reviewer comment: The numbers on the number of people living with HIV/AIDS needed to be reported along with the year, as stated in the first and second paragraphs of the introduction. At what time did this report occur?

Author response: The year of report was added

Reviewer comment: - In the introduction section, the authors noted that different countries use condoms at varying rates. To get a better general picture of condom usage among HIV/AIDS patients, I asked the authors to describe the prevalence of condom use among people living with the disease elsewhere in the world, with a focus on sub-Saharan Africa and Ethiopia.

Author response: We have added many literature across world with focusing on sub-Saharan Africa in revised manuscript

Reviewer comment: - be more appropriate to cite the prior comparable reviews and analyses that have concentrated on risky sexual activity, which is different from your work on regular condom use? Could you offer any alternative suggestions to the ones found in the earlier studies? On the other hand, tackling risky sexual behavior entails encouraging frequent condom usage, right? Could there be different results? I believe that all of your research and earlier studies focused on preventing HIV/AIDS and re-infection.

Author response: We have compared our study with many previous reviews in revised manuscript. There was a difference between risky sexual behavior and consistent condom use according to our study. 

Reviewer comment: - on objectives: The objective of the review was to conclude the magnitude and associated factors of consistent condom use among people living with HIV/AIDS in Ethiopia. It is better to say to detrmine instead of to conclude.

Author response: We have correct it as per your comments

Reviewer comment: - on search strategies: It would be best practice to include the search methods and the quantity of results for each database as an appendix or supplemental material. However, the authors either didn't do that or we don't have access to the supplemental table outlining the search methods and findings for each database.

Author response: We have included the search methods as supplemental material in previous revision

Reviewer comment: - Why a publication year-based subgroup analysis? Does it affect the study in any way?

Author response: The study only differ by publication year and region. The publication year can affects the outcome

Reviewer comment: - The discussion part still requires improvement and a more thorough comparison with past similar reviews and meta-analyses conducted worldwide. The comparison between this pooled prevalence and other studies' point prevalence was the authors' sole focus. When comparing the pooled data with point prevalence, is it good?

Author response: We have revised our discussion and it was compared with the reviews and meta-analyses conducted worldwide. We have compared the pooled data with the pooled data in majority of the cases. However, a few comparison was used for point prevalence in the case of shortage of literatures.

Reviewer comment: - On the other hand, the authors calculated the pooled odds ratio of factors affecting consistently using condoms in Ethiopia. then continue comparing the findings to those of another study done in Ethiopia (Dessie)? What caused it? What's the point? They neglected to include this study in their evaluation.

Author response: We have mistakenly compared with the study done in Ethiopia (Dessie).We have included the study in our analysis.

Reviewer comment: - In relation to the study's implications. Pharaphrasing and citations are necessary.

Author response: We have cited and paraphrased it

Reviewer comment: - The authors needed to describe the sub-group analysis by region in the discussion section for better clarity, policy implications, and to identify the region of requirements (regional variance). can explain the differences as well as make recommendations for areas with low prevalence?

Author response: We have discussed the difference in the consistent condom use across a different regions. We also put a recommendation for lower prevalence.

Reviewer comment:- In the limitation section? Does your study represent the whole region of Ethiopia? Are all regions of Ethiopia represented?

Author response: We have put as limitation that all regions of Ethiopia were not included

Reviewer comment:- The conclusion is far too short. Try to address all of the key findings and make workable suggestions.

Author response: We have added some important findings in conclusion part

Reviewer 2:

Reviewer comment: Title; Change peoples to people

Author response: We have changed it

Reviewer comment: Abstract: More than one million peoples…. (change peoples to people)

….among HIV patients…., change it to people living with HIV or people with HIV

We have used four databases such as PubMed, when you say such as is like you are giving examples of the databases but not exact databases you searched information. Revise this sentence because those databases are the one you sought information

Author response: We have changed the peoples to people, among HIV patients is change to people living with HIV, the word such as is like is replaced by word explaining the exact data base 

Reviewer comment: Main text

Methods: …Accordingly, the articles was (should be article was or articles were)

Results …majority of the participant was female in all articles ( should be majority of participants were..)

….who were disclosed their status was 5.61 time … (should be who disclosed their status…)

…patients who were live in urban area was 3.46 time (should be … who live in urban area were 3.46 times … or who are living in urban area were …..)

…patients who were married was 67% times ( should be patients who were married were 67% times…)

Author response: We have incorporated all your comments in revised version

Reviewer comment: In addition, the measure of effect used was odd ratio, however, the interpretation is like risk ratio.

It should be something like people living in urban areas have higher odds or reporting consistent condom use compared to rural areas. Not saying people living in urban areas are more likely reporting consistent condom use compared to rural areas.

Author response: We have described the associated factors by odds ratio in revised manuscript

Reviewer 3: 

Reviewer comment: Thank you for addressing the points highlighted in the previous review. The manuscript reads well and is technically sound.

Author response: Thank you for your comment

Thanks for your time and consideration,

 Regards!

---

## [Editor Report · Decision Letter 2]

9 May 2024

Magnitude of consistent condom use and associated factors among people living with HIV/AIDS in Ethiopia: Implication for reducing infections and re-infection. A systematic review and meta-analysis

PONE-D-23-16392R2

Dear Dr. Firomsa Bekele, MSc

We’re pleased to inform you that your manuscript has been judged scientifically suitable for publication and will be formally accepted for publication once it meets all outstanding technical requirements.

Kind regards,

Yimam Getaneh Misganie, PhD

Academic Editor

PLOS ONE

Additional Editor Comments:

o Contextualization: Provide context for the prevalence of consistent condom use in Ethiopia. How does it compare to other regions or global estimates? And Discuss the sociocultural factors that may influence condom use in the Ethiopian context.

o References: Ensure that all references adhere to the PLOS ONE citation style. Cross-check DOI numbers, journal titles, and page numbers for accuracy.

o Policy and Practice Implications: Highlight practical implications for public health programs. How can the findings inform interventions to promote consistent condom use? Remember to provide specific examples and actionable suggestions to help the authors improve their manuscript.

---

## [Editor Report · Acceptance letter]

21 Jun 2024

PONE-D-23-16392R2 

PLOS ONE

Dear Dr. Bekele , 

I'm pleased to inform you that your manuscript has been deemed suitable for publication in PLOS ONE. Congratulations! Your manuscript is now being handed over to our production team.

Kind regards, 

on behalf of

Dr. Yimam Getaneh Misganie 

Academic Editor

PLOS ONE